# Maintaining oxygen delivery is crucial to prevent intestinal ischemia in critical ill patients

Jochen J. Schoettler[1][☉]*, Thomas Kirschning[2][☉], Michael Hagmann[3], Bianka Hahn[1], Anna-Meagan Fairley[1], Franz-Simon Centner[1], Verena Schneider-Lindner[1,4], Florian Herrle[5], Emmanouil Tzatzarakis[5], Manfred Thiel[1], Joerg Krebs[1]

1 Department of Anaesthesiology and Surgical Intensive Care Medicine, University Medical Center Mannheim, Medical Faculty Mannheim of Heidelberg University, Mannheim, Germany, 2 Clinic for Thorax- and Cardiovascular Surgery HDZ NRW, University of Ruhr-University Bochum, Bochum, Germany, 3 Interdisciplinary Center for Scientific Computing, Heidelberg University, Heidelberg, Germany, 4 Department of Community Health Sciences, University of Manitoba, Winnipeg, Canada, 5 Surgical Department, University Medical Center Mannheim, University Medical Center Mannheim, Medical Faculty Mannheim of Heidelberg University, Mannheim, Germany

☉ These authors contributed equally to this work.
* jochen.schoettler@umm.de

## Abstract

### Background

Intestinal ischemia is a common complication with obscure pathophysiology in critically ill patients. Since insufficient delivery of oxygen is discussed, we investigated the influence of oxygen delivery, hemoglobin, arterial oxygen saturation, cardiac index and the systemic vascular resistance index on the development of intestinal ischemia. Furthermore, we evaluated the predictive power of elevated lactate levels for the diagnosis of intestinal ischemia.

### Methods

In a retrospective case-control study data (mean oxygen delivery, minimum oxygen delivery, systemic vascular resistance index) of critical ill patients from 02/2009–07/2017 were analyzed using a proportional hazard model. General model fit and linearity were tested by likelihood ratio tests. The components of oxygen delivery (hemoglobin, arterial oxygen saturation and cardiac index) were individually tested in models.

### Results

59 out of 874 patients developed intestinal ischemia. A mean oxygen delivery less than 250ml/min/m$^2$ (LRT vs. null model: p = 0.018; LRT for non-linearity: p = 0.012) as well as a minimum oxygen delivery less than 400ml/min/m$^2$ (LRT vs null model: p = 0.016; LRT for linearity: p = 0.019) were associated with increased risk of the development of intestinal ischemia. We found no significant influence of hemoglobin, arterial oxygen saturation, cardiac index or systemic vascular resistance index. Receiver operating characteristics analysis for

**Data Availability Statement:** All relevant data are within the manuscript and its Supporting Information files.

**Funding:** This study was supported by the foundation Klaus Tschira Stiftung. The funding source had no role in the design and conduct of the study.

**Competing interests:** The authors have declared that no competing interests exist.

**Abbreviations:** AUROC, Area under the Receiver Operating Characteristic; CI, cardiac output index; d, day; $DO_2I$, oxygen delivery index; h, hour; Hb, hemoglobin; IAP, intra-abdominal pressure; ICCA, Philips Intelli Space Crital Care and Anesthesia; ICD, International Classification of Diseases; ICIP, Philips IntelliVue Clinical Information Portfolio; ICU, intensive care unit; II, intestinal ischemia; LOS, length of stay; LRT, likelihood ratio test; $meanDO_2I$, mean oxygen delivery index; $minDO_2I$, minimal oxygen delivery index; OPS, Operation- and Procedures-code; PDMS, patient data management system; ROC, Receiver Operating Characteristic; $SaO_2$, arterial oxygen saturation; SAPS II, Simplified Acute Physiology Score II; SAS, Statistical Analysis System; $ScvO_2$, central venous saturation; SVRI, systemic vascular resistance index; y, year.

elevated lactate levels, pH, $CO_2$ and central venous saturation was poor with an area under the receiver operating characteristic of 0.5324, 0.52, 0.6017 and 0.6786.

## Conclusion

There was a significant correlation for mean and minimum oxygen delivery with the incidence of intestinal ischemia for values below 250ml/min/m$^2$ respectively 400ml/min/m$^2$. Neither hemoglobin, arterial oxygen saturation, cardiac index, systemic vascular resistance index nor elevated lactate levels could be identified as individual risk factors.

## Introduction

Intestinal ischemia (II) in critically ill patients is a life-threatening complication, leading to sepsis [1–3] caused by bacterial translocation [4–6] or direct fecal contamination of the peritoneal cavity. Congestive heart failure, diabetes mellitus, peripheral artery occlusive disease and age older than 60 years are recognized risk factors [7–9].

The mortality in these patients is increased [10–12] and it is one of the major missed diagnoses in deceased patients treated in intensive care units (ICU), implying an even higher incidence [13–15]. Overall mortality is estimated between 50% to 80% [1–3].

Surgical treatment within 24 hours (h) of diagnosis of II was identified as an independent predictor of survival, emphasizing the need for reliable risk stratification, specific markers, early detection and multidisciplinary management [16].

Acute obstruction with or without previously stenotic arterial vessels, mesenteric venous thrombosis and non-occlusive mesenteric ischemia because of impaired regional oxygen delivery are described as distinct pathophysiological entities leading to II. Although the splanchnic circulation receives approximately 20% of the cardiac output, several mechanisms like increased oxygen extraction and vascular autoregulation protect the intestines from ischemia. Nevertheless, a substantial reduction in oxygen delivery ($DO_2I$ in l/min/m$^2$) can lead to an imbalance between oxygen supply and demand and thereby cause II [7, 17–19]. The latter is of special interest for the intensivist as it might be preventable as insufficient $DO_2I$ due to low cardiac output (cardiac index, CI in l/min/m$^2$) combined with mesenteric and systemic vasoconstriction (systemic vascular resistance index, SVRI) caused by endo- or exogenous catecholamines leading to insufficient locoregional oxygen supply [1, 2, 9, 12, 20, 21].

Lactic acid, as well as pH, $CO_2$ and central venous saturation ($ScvO_2$), as routine parameters measured in the management of critically ill patients, are commonly used in a clinical setting to detect parenchymatous hypoxia, but the specificity for II is unknown. Furthermore, II per se induces lactic acid accumulation through parenchyma breakdown, as well as further pathological changes in the routine parameters.

Thus, the primary aim of this study is:

I. to define specific critical cut-off values for short term (the minimal $DO_2I$ during the 72 hours period before the diagnosis of II, $minDO_2I$) or prolonged oxygen delivery (the mean $DO_2I$ during the 72 hours period before the diagnosis of II, $meanDO_2I$) in the development of II

Secondary objectives of this study are:

I. to identify the role of the independent parameters (hemoglobin, Hb; arterial oxygen saturation, $SaO_2$ and CI) of $DO_2I$ in the development of II;

II. to identify the independent parameter (either Hb, $SaO_2$ or CI) whose manipulation is most beneficial in order to increase $DO_2I$ to a noncritical value to reduce the risk of II;

III. to evaluate the predictive power of elevated lactate levels, pH, $CO_2$ and central venous saturation ($ScvO_2$) for the diagnosis of II;

IV. to identify the influence of high SVRI on the development of II

## Material and methods

### Study design

This study was approved by the local ethics committee (Medizinische Ethikkommission II, University Medical Centre Mannheim, Medical Faculty Mannheim of the University of Heidelberg, Mannheim) (registration number 2016-800R-MA). The study was also registered at the Deutsche Register für klinische Studien (ID: DRKS00016030).

For this retrospective observational, non-interventional, monocenter case-control study the need for informed consent was waived by the local ethics committee.

The study was conducted in the 25-bed ICU of the Department of Anaesthesiology and Critical Care Medicine, University Medical Centre Mannheim, Medical Faculty Mannheim of the University of Heidelberg.

Data were retrospectively analyzed. The inclusion period lasted from 02/2009 to 07/2017 with an average of 1869 patients per year.

All patients who stayed longer than 72 hours, were older than 18 years and had a complete electronic medical record for calculating $DO_2I$ values were included in the analysis.

Irreversible parenchymal ischemia is induced in a time frame between 6 and 12h of hypoxia in the intestinal vascular zone [12, 19, 22]. As the goal of this study was to evaluate the impact of hypoxemia on the development of II and to discriminate the diagnosis of II attributable to critical care management from the sequelae of underlying diseases associated with II and originated before ICU admission, we excluded patients with a length of stay (LOS) shorter than 72h to exclude patients with undiagnosed II at admission on the ICU.

Furthermore, patients were excluded if they were <18 years old and if the electronic medical records were incomplete for calculating $DO_2I$ values.

As the overall incidence of II is low [3, 23, 24] we opted to include commonly recognized factors and pre-existing conditions [1, 3, 9] for the stratification of the Cox proportional hazards model, that might predestine the patient for intestinal hypoxia in case of acute severe illness necessitating treatment on ICU. So, identified patients with an elevated baseline risk for II by the following criteria: 1) congestive heart failure, 2) diabetes mellitus, 3) peripheral artery occlusive disease, 4) age older than 60 years [7–9].

Patients who developed II after admission with an ICU stay of at least 72h were grouped in the cases group to ensure a suitable amount of collected data for analysis.

The diagnosis of II was confirmed by clinician validation of medical records when at least one of the following criteria was fulfilled:

I. suggestive radiological signs for ischemia

II. endoscopic proof of ischemia

III. II specified in the pathology report

IV.  obvious ischemia detected intraoperatively without resection because of futility and an
ICU stay of at least 72 hours before the surgical intervention

All other patients without the diagnosis of II during their treatment on ICU were included
in the control group, were managed according to the standard operation procedures of our
unit, and received radiological or endoscopic interventions respectively surgical interventions
as indicated.

## Collection of data

All data were collected through Philips IntelliVue Clinical Information Portfolio (ICIP) and
Philips Intelli Space Critical Care and Anesthesia (ICCA) System. $SaO_2$, Hb and lactic acid
were measured routinely using a blood gas analyzer (Radiometer ABL 800 Flex, Radiometer,
Willich, Germany).

According to the standard operation procedures all patients with impaired cardiopulmo-
nary function were managed with a triple-lumen central venous catheter. Additionally, a trans-
pulmonary thermodilution catheter (Pulsiocath™, Pulsion Medical Systems, Munich,
Germany) was utilized in patients, when indicated by the attending physician.

The Pulse Contour Cardiac Output monitor (PiCCOplus™, Pulsion Medical Systems,
Munich, Germany) was used for measuring CI and SVRI with routine calibrations around
every 8h, averaging three daily $DO_2I$-measurements.

$DO_2I$ was calculated using a simplified version of the standard formula:

$$DO_2I \ (ml/min/m^2) = CI \ (ml/min/m^2) \ x \ SaO_2 x \ Hb \ (g/dl) \ x \ 1.34 \ x \ 10 \qquad (Eq \ 1)$$

And for calculating the SVRI we used the following formula:

$$SVRI \ (dyn * s * cm^{-5} * m^2) = \left[ \frac{(mean \ arterial \ pressure - central \ venous \ pressure)}{CI} \right] \times 80 \quad (Eq \ 2)$$

In order to quantify an insufficient oxygen delivery index within 72h before the diagnosis, we
calculated mean$DO_2I$ during the stay in ICU as a surrogate for a longer lasting hypoxic status
and the min$DO_2I$ during the stay in ICU to capture shorter periods of hypoxia. Furthermore,
we calculated the mean CI, mean $SaO_2$, mean Hb and mean SVRI during the stay in ICU.

Lactate levels were collected in the case group 72h before the diagnosis of II and in the con-
trol group we collected all lactate values over the ICU-stay. A plasma lactate concentration of
2mmol/l or less was defined as normal finding as this represents the clearing capacity for lactic
acid in normal adults [25].

## Statistical analysis

Metric data is presented as mean ± standard deviation, categorical data as absolute frequency
(percentage). P-values were calculated using the t-test and Fisher's exact test.

Because some variables for the $DO_2I$ measurements were not synchronously recorded we
allowed an 8h synchronization window for all variables for $DO_2I$.

Patients in the II group were matched according to the timepoint of the diagnosis of II (in
hours) with patients in the control group and an equal LOS (in hours) without II. CI, $SaO_2$,
Hb, mean$DO_2I$, min$DO_2I$ and meanSVRI from the last 72h was recorded in the individual
patient with II and in all patients in the case group with a corresponding LOS.

A stratified (by baseline risk) Cox proportional hazard model with time dependent covari-
ates [26] was then applied to assess the relationship between these parameters and the develop-
ment of II [7, 8].

We allowed a nonlinear relationship between the regressor and the hazard to develop an II by the application of smoothed regression splines [27]. For each model we assessed the general model fit, the linearity and the non-linearity of the regressor function by appropriate likelihood ratio tests (LRT).

In order to assess the effect of the individual components of $DO_2I$ (see Eq 1) we augmented the former mean$DO_2I$ model by the individual components (mean Hb, mean $SaO_2$ or mean CI) to derive adjusted coefficients and compared them to the unadjusted coefficients derived from a model that contains only the component of $DO_2I$ (Hb, $SaO_2$ and CI). Again, data from the last 72h of patients in the II group were matched with all control patients who had an equal LOS as the case.

Furthermore, we compared the model fit of the augment model with the mean $DO_2I$ model and the model that contains only the component of $DO_2I$ as regressor by appropriate LRT to determine the relative importance of each independent component.

We further conducted a Receiver Operating Characteristic Curve (ROC) analysis–sensitivity, specificity and Area under the ROC (AUROC) for lactate, pH, $CO_2$, $ScvO_2$ and their predictive value for II.

Statistical analysis was performed with R 3.3.2 (The R foundation for Statistical Computing, Vienna, Austria) [28] and the survival package and SAS 9.4 (Statistical Analysis System) [29, 30].

A p-value $\leq 0.05$ was regarded as statistically significant. No adjustment for multiplicity was applied.

All dedicated statistician (MH) was responsible for the calculations.

## Results

From 02/2009 to 07/2017 we analyzed 15032 patients of whom 215 patients developed II during their ICU stay. 119 patients fulfilled the minimum required ICU stay of > 72h. 60 patients had to be excluded because no advanced hemodynamical monitoring was established. Thus, a total of 59 patients fulfilled all inclusion criteria (Fig 1). Baseline characteristics are presented in Table 1. We identified 33 female and 26 male II patients with an average age of 62.4 ± 14.6 years. Patients with II had a significantly higher Simplified Acute Physiology Score (SAPS II) score on admission than controls (47.3 ± 13.7 vs. 43.1 ± 13.1, p = 0.025). II was associated with a prolongation of the ICU stay (25.0 ± 22.2 vs. 18.5 ± 16.1, p = 0.032). ICU-mortality was higher in the II group (66.1%) compared to the control group (32.1%) (p < 0.0001). None of the evaluated comorbidities were significantly more prevalent in patients with II.

We found a significant non-linear influence of mean$DO_2I$ on ischemia hazard (LRT vs null model: $\chi^2$ (df = 3.23) = 10.536, p = 0.018; LRT for non-linearity: $\chi^2$ (df = 2.23) = 9.281, p = 0.012) (Fig 2A). The application of this model showed, that the relative ischemia hazard (reference: mean$DO_2I$ = 500ml/min/m$^2$) is significantly elevated when mean$DO_2I$ falls below approximately 250ml/min/m$^2$ and increases disproportionately with smaller values (Fig 2B).

We observed a qualitatively similar relationship between min$DO_2I$ and relative ischemia hazard (Fig 3). First, the influence of min$DO_2I$ was shown to be linear (LRT vs null model: $\chi^2$ (df = 1.39) = 6.8, p = 0.016; LRT for linearity: $\chi^2$ (df = 1.00) = 5.48, p = 0.019); LRT for non-linearity: $\chi^2$ (df = 0.39) = 1.103, p = 0.191), thus the increase for smaller values is less steep. Secondly, it was observed that the relative ischemia hazard is already significantly elevated at a min$DO_2I$ value of approximately 400ml/min/m$^2$ compared to the reference value.

Our assessment of the individual components of $DO_2I$ showed that no single component had a significant influence on the ischemia hazard no matter if we adjust for mean$DO_2I$ or not (Table 2). Consequently, we could not improve the model fit to our data of the mean$DO_2I$ model by adding individual components (all LRT p-values were at least 0.104), but we

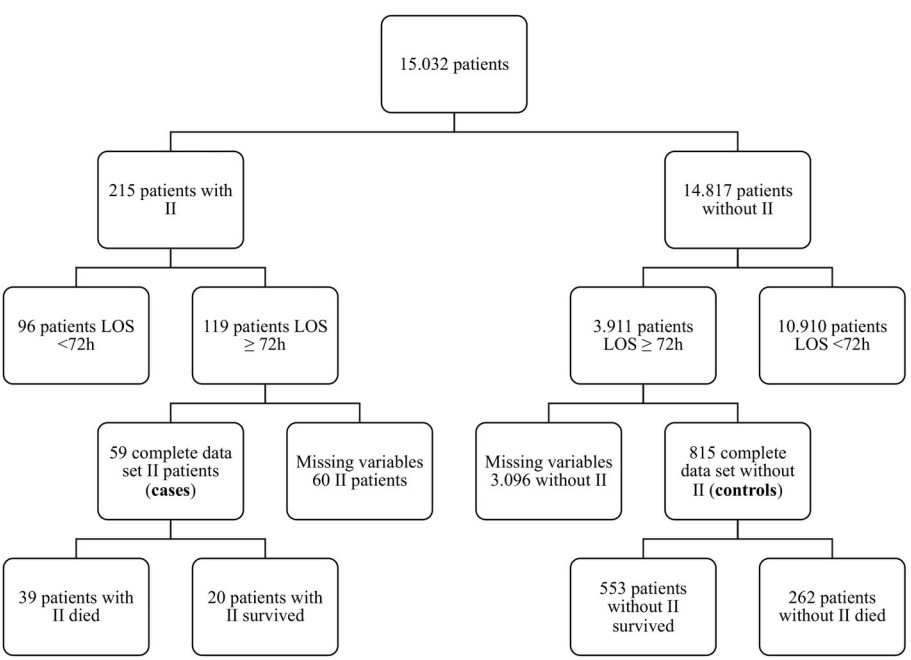

**Fig 1. Patient selection flow diagram.** II = intestinal ischemia, LOS = length of stay, h = hour.

observed a significantly poorer model fit when we dropped meanDO$_2$I from the model for each component (all LRT p-values were below 0.024).

Our analysis of lactate levels showed that 51 Patients with II had lactate levels $\geq$ 2mmol/l (86%), eight patients showed lactate levels $<$ 2mmol/l (14%) resulting in a sensitivity of 86.44%. In the control group there were 683 patients with lactate levels $\geq$ 2mmol/l (84%) and 132 (16%) patients with lactate levels $<$ 2mmol/l, leading to a specificity of 16.2%. Receiver

**Table 1. Patients baseline characteristics.**

| | Intestinal ischemia | Nonischemia controls | p-value |
|---|---|---|---|
| n | 59 | 815 | |
| Sex (m/f) | 26/33 | 494/321 | **0.0136** |
| Age (y) | 62.4 ± 14.6 | 60.7 ± 15.9 | 0.3766 |
| SAPS II (points) | 47.3 ± 13.7 | 43.1 ± 13.1 | **0.0247** |
| Length of stay (d) | 25.0 ± 22.2 | 18.5 ± 16.1 | **0.0316** |
| ICU-mortality | 39 (66.1%) | 262 (32.1%) | **<0.0001** |
| Congestive heart failure | 20 (33.9%) | 205 (25.2%) | 0.1642 |
| Diabetes mellitus | 25 (42.4%) | 378 (46.4%) | 0.5903 |
| Peripheral vascular occlusive disease | 8 (13.6%) | 56 (6.9%) | 0.0679 |
| Coronary heart disease | 8 (13.6%) | 155 (19.0%) | 0.3867 |
| COPD | 4 (6.8%) | 83 (10.2%) | 0.5041 |
| Artrial fibrillation | 29 (49.15%) | 297 (36.44%) | 0.069 |
| Chronic renal disease | 8 (13.56%) | 81 (9.94%) | 0.3714 |
| Nicotine abuse | 5 (8.47%) | 77 (9.45%) | 1.0 |

Patients baseline characteristics; n = number of patients, m = male, f = female, y = year, SAPS II = simplified acute physiology score II, d = day, ICU = intensive care unit, COPD = chronic obstructive pulmonary disease

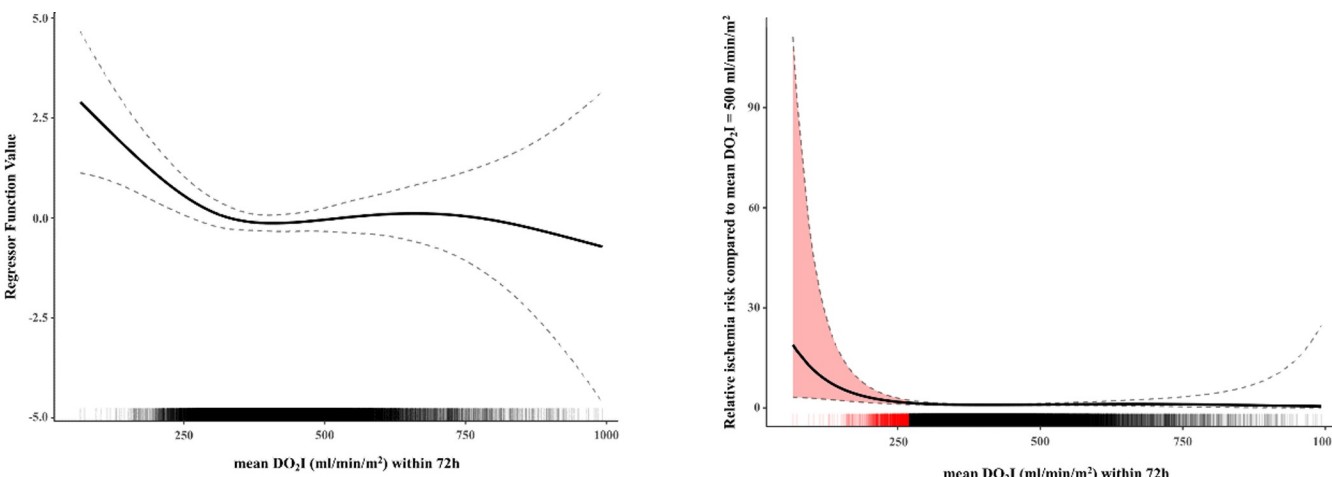

**Fig 2.** A. Regressor plot of meanDO$_2$I within 72h. DO$_2$I = oxygen delivery index, h = hour, the solid line shows the regressor plot for meanDO$_2$I, dashed lines show the standard deviation, above the x-axis the 14.320 DO$_2$I calculations are plotted as single small lines. B. Relative ischemia risk compared to mean DO$_2$I within 72h before the onset of intestinal ischemia. Y-axis shows increasing relative ischemia risk with decreasing meanDO$_2$I values (x-axis, solid line) by DO$_2$I values below approximately 250ml/min/m$^2$ (highlighted by the red marked area), dashed lines show the standard deviation, above the x-axis the 14.320 DO$_2$I calculations are plotted as single small lines (critical DO$_2$I values are highlighted as a red line), DO$_2$I = oxygen delivery index, h = hours.

operating characteristics analysis was poor with an Area under the ROC of 0.5324. The analysis of pH also showed a very high sensitivity with 90%, but a very low specificity with 21%. Resulting in a poor receiver operating characteristics analysis with an area under the ROC of 0.52. Furthermore, the variables pCO$_2$ and ScvO$_2$ also did not perform well with a sensitivity of 83% and a specificity of 38%, with a resulting ROC analysis of 0.6017 for CO$_2$ and 77% sensitivity, specificity of 52%, and AUROC of 0.6786 for ScvO$_2$, respectively (Fig 4).

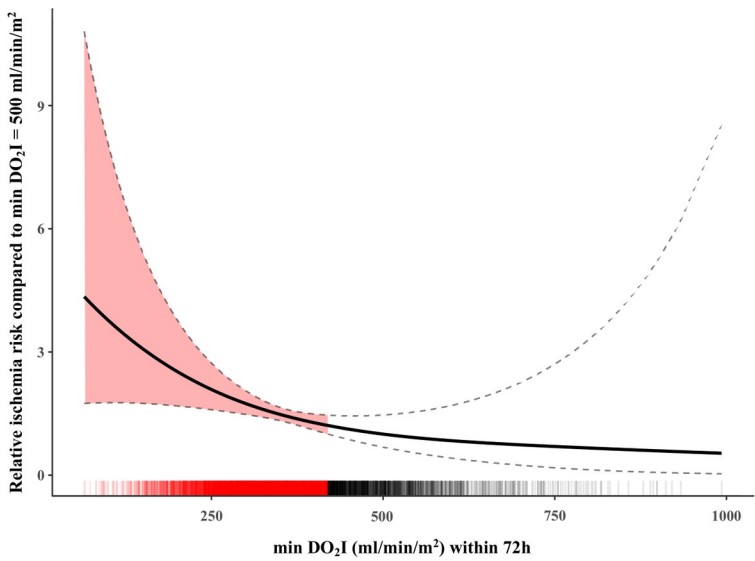

**Fig 3. Relative ischemia risk compared to min DO$_2$I within 72h before the onset of intestinal ischemia.** Y-axis shows increasing relative ischemia risk with lower minDO$_2$I values (x-axis, solid line) by DO$_2$I values below approximately 400ml/min/m$^2$ (highlighted by the red marked area), dashed lines show the standard deviation, above the x-axis the 14.320 DO$_2$I calculations are plotted as single small lines (critical DO$_2$I values are highlighted as a red line), DO$_2$I = oxygen delivery index, h = hours.

**Table 2. Individual component of DO₂I.**

| Component | Unadjusted | | | | Adjusted | | | |
|---|---|---|---|---|---|---|---|---|
| | HR | Coef | SD | p-value | HR | Coef | SD | p-value |
| CI | 0.838 | -0.177 | 0.154 | 0.25 | 0.997 | -0.003 | 0.277 | 0.992 |
| SaO₂ | 0.90 | -0.108 | 0.065 | 0.1 | 0.885 | -0.122 | 0.066 | 0.063 |
| Hb | 1.001 | 0.001 | 0.101 | 0.994 | 1.030 | 0.03 | 0.111 | 0.787 |

Individual component of $DO_2I$; HR = hazard ratio, coef = coefficient, SD = standard deviation, p = p-value, CI = cardiac output index, $SaO_2$ = arterial oxygen saturation, Hb = hemoglobin

SVRI had no significant association with ischemia hazard (LRT vs null model: $\chi^2$ (df = 1.41) = 2.522, p = 0.178) (Table 3).

## Discussion

The findings in this study were:

I. There is a critical cut-off value for $meanDO_2I$ of approximately 250ml/min/m² representing a longer state of insufficient oxygen delivery in a 72h timeframe before the clinical diagnosis of II which is affecting the risk for the development of II. Lower $meanDO_2I$ values strongly increase risk of II.

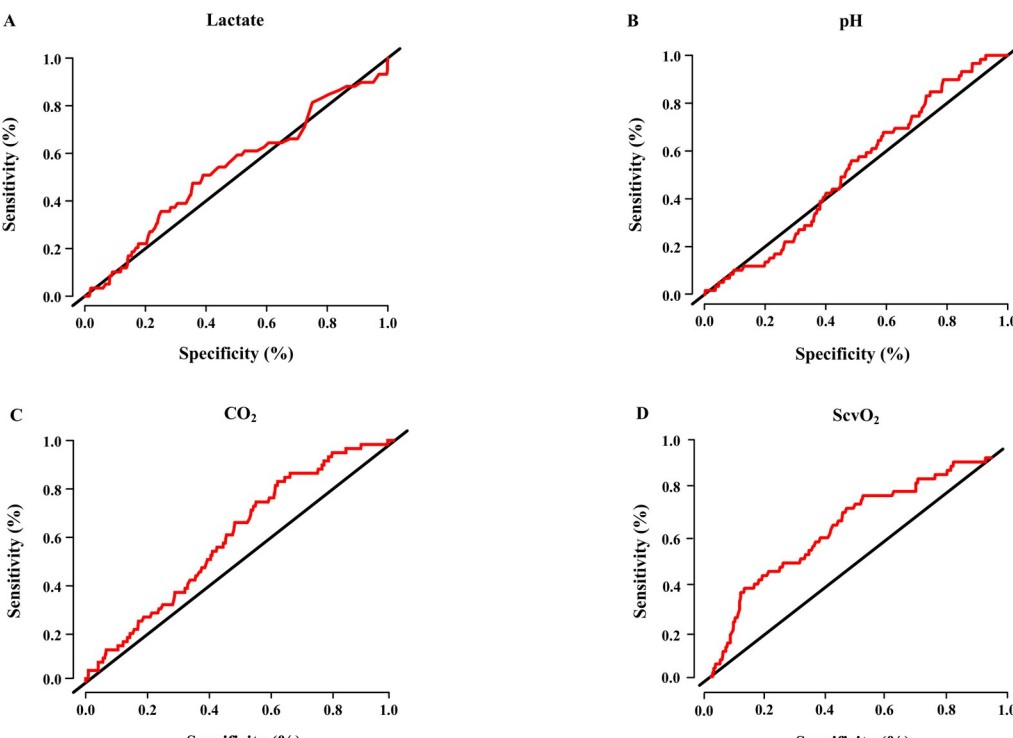

**Fig 4. Receiver operating characteristics analysis for lactate, pH, CO₂, ScvO₂ and the development of intestinal ischemia.** Panel A lactate: Area under the ROC curve: 0.5324; sensitivity 86.44% and specificity 16.2% for a cut-off value of lactate $\geq$ 2mmol/l; Panel B pH: Area under the ROC curve: 0.52; sensitivity 90% and specificity 21%; Panel C $CO_2$: Area under the ROC curve: 0.6017; sensitivity 83% and specificity 38%; Panel D $ScvO_2$: Area under the ROC curve: 06786.; sensitivity 77% and specificity 52%; ROC = Receiver operating characteristic.

**Table 3. Influence of systemic vascular resistance.**

| test | chisq | df | p-value |
|------|-------|-----|---------|
| null model | 2.522 | 1.414 | 0.178 |
| linear | 1.400 | 1.000 | 0.237 |
| non-linear | 0.641 | 0.414 | 0.219 |

Influence of systemic vascular resistance; Chisq = Chi-squared test, df = degrees of freedom

II. Even a single $minDO_2I$ value smaller than $400ml/min/m^2$ increases the risk for II.

III. Neither Hb, $SaO_2$ nor CI as indiviual components of $DO_2I$ showed significant diagnostic superiority compared to $DO_2I$ in predicting II.

IV. Therapeutic decisions based on lactate, pH, $CO_2$ or $ScvO_2$ are not useful for the prevention or early detection of II especially because of their low specificity.

V. In our analysis SVRI had no effect on the incidence of II.

## Delivery of oxygen and survival

To our knowledge no prior study investigated whether there is a crucial $DO_2I$ cut off value for organ dysfunction like II. A $meanDO_2I$ of $250ml/min/m^2$ over 72h on ICU and a $minDO_2I$ value smaller than $400ml/min/m^2$ substantially increases the risk for developing II and could alert the attending clinician accordingly. The fact that $minDO_2I$ has an earlier effect on the development of II may be due to cellular compensatory mechanisms that have not yet been activated, in the sense of ischemic preconditioning.

Ischemic preconditioning reduces ischemia-reperfusion injury by inhibiting and reducing the inflammatory response in the reperfusion phase [31].

Single low $DO_2I$ events without ischemic preconditioning result in reduced mitochondrial ATP generation, as well as other pathological mechanisms. In the subsequent reperfusion, a pronounced inflammatory response occurs, which further exacerbates ischemia [31–33]. Guan et al. [34] showed using vivo microscopy that adverse effects due to short-term ischemia are partly completely reversible in the reperfusion phase. However, they showed that after prolonged periods of ischemia, normal cell structures and functions could not be fully restored in a large proportion of cells and during reperfusion further deterioration occurred.

In the present study, this microscopically proven pathology by Guan et al. [34] is also supported, as short-term low $DO_2I$ values have a markedly lower risk of ischemia than longer-lasting low $DO_2I$ ($meanDO_2I$) phases.

There are no established guide values for $DO_2I$ and oxygen consumption for critically ill patients. In a normal resting adult the normal $DO_2I$ is approximately $500ml/kg/m^2$ assuming a CI of $2500 ml/min/m^2$, a Hb of 15 g/dl and a $SaO_2$ of 100%, from which $125ml/min/m^2$ are consumed through the normal metabolism [35]. As a $DO_2I$ of $500ml/min/m^2$ is commonly reported as reference value in healthy subjects and typically not associated with II and post procedural complications it was chosen as a reference value [36, 37]. On top of that a $DO_2I$ of $500ml/min/m^2$ was tested as a safe endpoint for shock resuscitation [38].

Shoemaker et al. studied hemodynamic parameters, $DO_2I$ and oxygen consumption in critically ill patients, showing a correlation between less organ failure as well as survival and supranormal values of $DO_2I$, oxygen consumption and cardiac index [39–43]. The authors theorized that morbidity and mortality could be reduced in critically ill patients if these parameters were used as therapeutic goals.

Subsequently, controlled randomized trials investigated the effect of hemodynamic optimization in critically ill patients, manipulating $DO_2I$ to supranormal values [43–50]. Several of these studies [43, 47–51] showed a decrease in mortality and morbidity when $DO_2I$ was manipilated to a supranormal value before surgery and during the peri- and postoperative period.

A meta-analysis by Kern et al. [52] summarized relevant prospective randomized trials analyzing hemodynamic optimization in high-risk patients. In trials with hemodynamic optimization before the onset of organ dysfunction a significant reduction in mortality [47–51] could be demonstrated. Hemodynamic optimization after the onset of organ dysfunction however caused no related mortality reduction [44–46, 53]. Our data supports the idea that hemodynamic optimization before II became clinical apparent might prevent these complications in our cases. On the other hand, it remains speculative whether there is a significant risk reduction for II due to supranormal $DO_2I$ values.

## Effects of Hb, $SaO_2$ and CO on II

Relevant studies in the field utilized standardized protocols to keep $DO_2I$ values in normal or supranormal levels by manipulating all $DO_2I$ components depending on arbitrarily elected cut-off values for CI, Hb and $SaO_2$ [39–51, 53, 54]. None of them distinguished which component of $DO_2I$ is most effective to optimize. In this study we showed, that $DO_2I$ as a goal parameter for optimization might be relevant to prevent II, but the individual components seem equally important to prevent II.

## The role of lactate, pH, $CO_2$ and $ScvO_2$ in the diagnosis of II

Lactate is a well-known marker of parenchymatous hypoxia regularly reported in studies revising II [55–57] and its measurement is recommended in recent guidelines [3, 9]. On the other hand, many studies and meta-analyses confirm that the classical routine parameters are of no value in distinguishing patients with II from those without [9, 58, 59]. In a retrospective multicenter study by Leone et al. [60] investigating risk factors associated with ICU-mortality in patients with II, lactate levels higher than 2.7mmol/l were found to be an independent predictor for ICU-mortality. Yet the author pointed out that lactate is not a useful tool for diagnosis or exclusion of II, because of its low sensitivity and specificity. Bourcier et al. [61] investigated patients with suspected II and also collected lactate levels, showing no statistically significant difference between lactate levels of II patients and patients without II. In a prospective trial Murray et al. [62] found a significant elevation in D-lactate levels in patients with II compared to controls. Sensitivity and specificity were 90% and 87%. In summary, lactate not differentiated in its D- and L- enantiomers appears to be a good parameter for mortality estimation [60, 63] but not a reliable paramter for the diagnosis of II.

The goal of the study was to evaluate the correlation of $DO_2I$ and the development of II in patients treated on the ICU. So, we hypothesize a time dependent clinical inapparent sequence of clinical inapparent inadequate (locoregional) delivery of oxygen, inducing irreversible II and corresponding lactate accumulation leading to clinical detectable sequalae like vasomotor dysfunction and endothelial leak which then enable the clinician at the bedside to diagnose and manage II. We therefore wanted to connote our findings of a "cut-off" $DO_2I$ with the corresponding lactate levels. Our findings of relative high mean and min$DO_2I$ associated with II might help to explain the relative low sensitivity and specificity of lactate in the diagnosis of II.

Cruz et al. [64] reported an increase in the intestinal-arterial $pCO_2$ gradient in a model of small bowel ischemia-reperfusion that corresponded with the grade of the mucosal damage. In line with that finding, Siniscalchi et al. [65] found a significant lower pH and higher $PaCO_2$ in patients who underwent small bowel transplantation comparing their baseline measurement

and 120 minutes after reperfusion of the graft. The authors hypothesized that the fall of pH after the revascularization and the concomitant rise in $PaCO_2$ was noted due the increased metabolic activity in the new organ. We hypothesized that an increase in $PaCO_2$ or a reduction of pH, either due to lactic acidosis or due to transient reperfusion of underperfused intestinal organs might be a valuable parameter for II. As shown by our ROC analysis unfortunately neither pH nor $PaCO_2$ showed a clinically useful specificity for the prediction of II. This might be caused by the relative insensitivity of global changes in both parameters compared to direct measurements in the intestinal mucosa. On the other hand, in a substantial part of critical ill patients hypercapnia and the corresponding acidosis are caused by guideline-compliant management [66, 67] and not associated with intestinal ischemia at all. The measurement of the central venous oxygen saturation is discussed in the recent guidelines for the management of septic shock and represents global oxygen extraction and utilization in critically ill patients [68]. As shown by Heino et al. [69] the oxygen extraction in II is increased. Unfortunately, as shown in our ROC analysis corresponding changes in $ScvO_2$ lack the necessary specificity to represent a useful prognostic marker of II in a clinical setting. This finding might be caused by the dichotomy of the parameter regarding oxygen delivery. A low $ScvO_2$ is usually a sign of hypoxia or insufficient cardiac output, an increased $ScvO_2$ usually denotes an impaired oxygen extraction [70, 71].

## The role of SVRI in the development of intestinal ischemia

The hypothesis that endo- or exogenouos catecholamines may induce II because of reduced oxygen delivery to intraabdominal organs due to mesenteric vasoconstriction is proposed in many guidelines and trials [1, 3, 9, 72]. In this study we utilized systemic vascular resistance index as surrogate for vasoconstriction irrespectively of exogenouos catecholamines. We found no significant correlation between SVRI and the incidence of II. It should be noted, as SVRI is the result of the physiological effects of endo- or exogenouos catecholamines, our analysis is indepented of the catechoalmine therapy and other therapeutical decisions of the attending physicians.

## Limitations

Results of this study were potential biased due to a different pre-existing disease profile, the heterogeneity of medical history and the clinical course, for example, new or different comorbidities in the investigated cohort. As it was the goal of this study to evaluate the effects of the delivery of oxygen and its independent parameters (Hb, arterial $SaO_2$ and CI) on the development of II we tried to attenuate these factors by identifying and adjusting for anamenstic proxies and conditions for a higher risk of II like diabetes mellitus, peripheral vascular disease, chronic heart failure, coronary heart and pulmonary diseases and used them for the stratification of the Cox model. Therefore we opted not to include factors like prognostic scores evaluating physiological criteria like APACHE II or SOFA score in the Cox model as they reflect acute severity of illness of the patient and not necessarily predestine the patient for II per se.

As we had no opportunity to acquire advanced hemodynamic data like cardiac output from the patients included in this retrospective study prior to admission to the ICU we explicitly excluded patients with a length of stay shorter than 72h from the study. Naturally we suspect that patients suffering II prior to admission on the ICU might present a significant lower $meanDO_2I$ and $minDO_2I$ then the II group in this study. Therefore, we acknowledge that we probably evaluated a distinct subgroup of patients suffering II and our findings cannot be extrapolated to all patients with II.

Furthermore we did not account for therapeutic interventions to manage II once the clinical diagnosis was made.

Lastly, in the analysed cohort, surgical patients with abdominal pre-existing conditions might have biased the results.

## Conclusion

To our knowledge, this is the first study to show a direct correlation between the incidence of II and a critical $DO_2I$ value. Our findings emphasize the need to keep $DO_2I$ at an adequate level to prevent deterioration of the patients condition as well as their outcome due to the development of II. This crucial cut-off value for $DO_2I$ may enable intensive care physicians to identify patients at risk and also allow for optimization of therapy by manipulation of the $DO_2I$ parameters to prevent II.

## Supporting information

**S1 Dataset. Anonymized data set.**
(XLSX)

**S2 Dataset. Annotation for the anonymized data set.**
(XLSX)

## Acknowledgments

The authors would like to thank the staff of the intensive care unit for their support.

## Author Contributions

**Conceptualization:** Jochen J. Schoettler, Thomas Kirschning, Manfred Thiel, Joerg Krebs.

**Data curation:** Jochen J. Schoettler, Michael Hagmann, Bianka Hahn, Franz-Simon Centner.

**Formal analysis:** Jochen J. Schoettler, Michael Hagmann, Bianka Hahn.

**Supervision:** Manfred Thiel, Joerg Krebs.

**Writing – original draft:** Jochen J. Schoettler, Joerg Krebs.

**Writing – review & editing:** Anna-Meagan Fairley, Franz-Simon Centner, Verena Schneider-Lindner, Florian Herrle, Emmanouil Tzatzarakis, Manfred Thiel, Joerg Krebs.

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
