## [Decision Letter · Decision Letter 0]

9 Apr 2021

PONE-D-21-08068

Maintaining oxygen delivery is crucial to prevent intestinal ischemia in critical ill patients

PLOS ONE

Dear Dr. Schöttler,

Thank you for submitting your manuscript to PLOS ONE. After careful consideration, we feel that it has merit but does not fully meet PLOS ONE’s publication criteria as it currently stands. Therefore, we invite you to submit a revised version of the manuscript that addresses the points raised during the review process.

We look forward to receiving your revised manuscript.

Kind regards,

Corstiaan den Uil

Academic Editor

PLOS ONE

Journal Requirements:

Reviewers' comments:

Reviewer's Responses to Questions

**Comments to the Author**

1. Is the manuscript technically sound, and do the data support the conclusions?

Reviewer #1: Partly

2. Has the statistical analysis been performed appropriately and rigorously? 

Reviewer #1: No

3. Have the authors made all data underlying the findings in their manuscript fully available?

Reviewer #1: No

4. Is the manuscript presented in an intelligible fashion and written in standard English?

Reviewer #1: Yes

5. Review Comments to the Author

Reviewer #1: The work analyzes several hemodynamic and perfusion clinical variables used to evaluate the supply of oxygen at a systemic level, seeking to prevent intestinal ischemia in critically ill patients. Gastrointestinal dysfunction is one of the main causes of morbidity and mortality in critically ill patients. It is a good retrospective study; however, I have some comments on the manuscript that need to be clarified or improved.

In the introduction, they talk about the complications of intestinal ischemia and its incidence, however the bibliography used is not up-to-date. They talk about mortality percentages and the lack of understanding of the pathophysiology of intestinal ischemia using a 2004 work as a reference. There are works from the last 5 years on the subject that they could refer to. There are several publications on gastrointestinal dysfunction, Ischemia / reperfusion damage, which could be referenced.

I think the study should be better founded, since several things that they mention about the variables they analyzed have already been studied and reported in critically ill patients and in various models of gastrointestinal ischemia.

Methodology

What was the basis for defining the criteria used to define high risk of II? They were separated for being patients with cardiovascular problems and over 60 years of age. Why wasn't a criterion used like a certain score value on a scale like SOFA or APACHE? The criteria that they defined, in what way did they apply it?

Why were patients with II on admission or who developed it in the first 72 hours in the ICU excluded from your study? It would have been interesting to evaluate his hemodynamic variables compared to what was observed days after admission to the ICU.

How did you define the inclusion criteria for patients in the control group? In that group were there radiological studies, endoscopy or any pathology report that showed that they had no gastrointestinal involvement?

Why was the DO2 not calculated based on LOS? If they have a lot of variability in the number of days of stay of the patients? They have patients of 3 days compared to patients of more than a month of stay.

Nor does it seem appropriate to me that the variables of CI, SaO2, Hb and SVRI were averaged during the ICU stay with such variability in the days. Especially considering that these variables change rapidly in ICU according to the treatments.

Why only analyze lactate, and not PCO2 or arterial and venous pH if you had the data?

Results

The groups are very unbalanced, therefore the analyzes carried out have statistical significance when, in fact, due to the variability of the data, in groups with the same number of data, they would not present significant changes. Why not use some criteria to select the patients from the control group with whom the study cases are compared? If you have 59 cases of II vs. 815 as controls, why is there so much dispersion in the controls for all the variables analyzed?

The figures have very poor quality, I suggest improving them all.

Why use mean DO2I of 500 ml / min / m2 as a reference? If ICU patients normally have lower values?

Regarding the lactate analysis, they have higher levels in the control group, if the n of each group are considered, for which I do not consider the ROC analysis they present adequate. Why not use venous oxygen saturation which reflects systemic oxygenation?

There are studies reported in critically ill patients with ischemia showing that lactate is not a good marker. The same authors report some references related to its low sensitivity and specificity. So why did they use it to predict II?

I believe that the discussion should better relate the results of this work with what was previously reported. Why not analyze gastrointestinal ischemia or ischemia / reperfusion studies and compare their findings if they are closely related to what they are presenting? Many of the ICU patients develop sepsis, did they record the comorbidities of the patients included in the study?

It's interesting work, but I think the information and data analysis could be better presented.

6. PLOS authors have the option to publish the peer review history of their article (what does this mean?). If published, this will include your full peer review and any attached files.

Reviewer #1: No

---

## [Author Response · Author response to Decision Letter 0]

21 Jun 2021

Reviewer comments to the Author:

1. In the introduction, they talk about the complications of intestinal ischemia and its incidence, however the bibliography used is not up-to-date. They talk about mortality percentages and the lack of understanding of the pathophysiology of intestinal ischemia using a 2004 work as a reference. There are works from the last 5 years on the subject that they could refer to. There are several publications on gastrointestinal dysfunction, Ischemia / reperfusion damage, which could be referenced.

I think the study should be better founded, since several things that they mention about the variables they analyzed have already been studied and reported in critically ill patients and in various models of gastrointestinal ischemia.

Re.: We thank the reviewer for addressing this important issue. We updated the references accordingly (citations 2, 3, 7, 8, 12, 17, 18, 19 and 21 in the manuscript). Furthermore we precised the pathophysiological rationale of the study in the introduction (pages 3 and 4, lines 65-76).

2. What was the basis for defining the criteria used to define high risk of II? They were separated for being patients with cardiovascular problems and over 60 years of age. Why wasn't a criterion used like a certain score value on a scale like SOFA or APACHE? The criteria that they defined, in what way did they apply it?

Re.: This is a very important point and we would like to thank the Reviewer for this comment. The stratified Cox proportional hazards model we used to study the effect of various parameters on the instantaneous hazard experienced by individuals. For the stratification of the model, risk factors described in the literature (e.g. congestive heart failure, diabetes mellitus, peripheral artery occlusive disease and age older than 60 years) were considered and inserted into the model [1, 2]. As the overall incidence of intestinal ischemia is low [3-5] we opted to include commonly recognized factors and pre-existing conditions [5-7] in the hazards model, that might predestine the patient for intestinal hypoxia in case of acute severe illness necessitating treatment on ICU. We specified this in the materials section of the manuscript (pages 5 and 6, lines 119-124).

Contrary to our knowledge prognostic scores evaluating acute physiological criteria like APACHE II or SOFA score are not recognized risk factors for the development of intestinal ischemia. As it was the goal of this study to evaluate the effects of the delivery of oxygen and its independent parameters (hemoglobin, arterial oxygen saturation and cardiac output) on the development of intestinal ischemia and not necessarily the acute severity of illness of the patient, acute physiological scores were not included in the stratification of the Cox proportional hazards model. We present the SAPS II score at admission of both groups in table 1 and included this point in the limitations (pages 19, lines 423-431).

3. Why were patients with II on admission or who developed it in the first 72 hours in the ICU excluded from your study? It would have been interesting to evaluate his hemodynamic variables compared to what was observed days after admission to the ICU.

Re.: This is an important point and merits careful consideration. The goal of the study was to evaluate the effects of the delivery of oxygen respectively hemoglobin, arterial oxygen saturation and cardiac output on the development of intestinal ischemia. Furthermore, we wanted to elaborate on the relevance of lactate, pH, PaCO2 and ScvO2 as indicators for parenchymal ischemia. As we had no opportunity to acquire advanced hemodynamic data like cardiac output from the patients included in this retrospective study prior to admission to the ICU, we explicitly excluded patients with a length of stay shorter than 72 hours from the study. Naturally we suspect that patients suffering II prior to admission on the ICU might present a significant lower meanDO2I and minDO2I then calculated in the II group in this study. Therefore, we acknowledge that we probably evaluated a distinct subgroup of patients suffering II and our findings cannot be extrapolated to all patients with II. We clarified this fact in the Material and Methods section (page 5, line 111-118) and discussed it in the limitations section of the manuscript (page 19, lines 432-438).

4. How did you define the inclusion criteria for patients in the control group? In that group were there radiological studies, endoscopy or any pathology report that showed that they had no gastrointestinal involvement?

Re.: We would like to thank the Reviewer for this important comment. For the control group all patients were included when they were older than 18 years and the ICU-stay was longer than 72 hours and there was no clinical diagnosis of II. Furthermore, all patients included in the study were managed according to the standard operation procedures of our unit and received radiological or endoscopic interventions respectively surgical interventions as indicated (page 6, line 136-139).

5. Why was the DO2 not calculated based on LOS? If they have a lot of variability in the number of days of stay of the patients? They have patients of 3 days compared to patients of more than a month of stay.

Re.: Patients in the II group were matched according to the timepoint of the diagnosis of II (in hours) with patients in the control group and an equal LOS (in hours) without II. CI, SaO2, Hb, meanDO2I, minDO2I and meanSVRI from the last 72 hours was recorded in the individual patient with II and in all patients in the case group with a corresponding LOS. 

A stratified (by baseline risk) Cox proportional hazard model with time dependent covariates [8] was then applied to assess the relationship between these parameters and the development of II.

We also represented this fact graphically, by showing the number of DO2I measurements below the figures 2A, 2B and 3, respectively by means of dashes (page 11 lines 236-237, page 11 lines 242-244 and pages 11 and 12 lines 256-258). The more dashes are shown, the more DO2I calculations were performed, resulting in a total 14.320 single DO2I measurements for analysis. We clarified this in the Material and Methods section (page 8, lines 174-180 and lines 190-192).

6. Nor does it seem appropriate to me that the variables of CI, SaO2, Hb and SVRI were averaged during the ICU stay with such variability in the days. Especially considering that these variables change rapidly in ICU according to the treatments.

Re.: This is a very correct statement and we thank the reviewer for the chance to describe this point more detailed. 

This is a correct statement and we thank the reviewer for the chance to describe this point in more detail. 

Again, the length of stay was considered in the analysis. First the timepoint of intestinal ischemia was detected in the case group and subsequently the last 72 hours were reviewed. Within these 72 hours meanCI, minSaO2, meanHb and meanSVRI were calculated in patients with II and in the corresponding cases without II and an equal LOS.

For the control group, all patients were considered who had the same length of stay and no intestinal ischemia at this timepoint. Then meanDO2I, minDO2I and SVRI were also calculated retrospectively for 72 hours. This allowed to take the possible rapid changes in CI, SaO2, Hb and SVRI into account.

7. Why only analyze lactate, and not PCO2 or arterial and venous pH if you had the data?

Re.: According to the recommendations of the reviewer we included ROC analysis of pH, paCO2 and ScvO2 in the manuscript and discussed their value for the prediction of II in the context of our retrospective study (discussion pages 17 and 18, lines 387-408).

8. The groups are very unbalanced, therefore the analyzes carried out have statistical significance when, in fact, due to the variability of the data, in groups with the same number of data, they would not present significant changes. Why not use some criteria to select the patients from the control group with whom the study cases are compared? If you have 59 cases of II vs. 815 as controls, why is there so much dispersion in the controls for all the variables analyzed?

Re.: We fully agree with the reviewer regarding the large difference in the patients included in the II and control group reflecting the low incidence of II and potentially compromising the analysis. Patients in the II group were matched according to the timepoint of the diagnosis of II (in hours) with patients in the control group and an equal LOS (in hours) without II.

We specified this in the Materials and Methods section (page 8, lines 174-180 and lines 190-192). The project was accompanied by a dedicated statistician who is one of the authors of the manuscript (M.H.) to ensure comparability of the case and control groups.

9. The figures have very poor quality, I suggest improving them all.

Re.: We improved the quality of the figures accordingly to the recommendations of the reviewer.

10. Why use mean DO2I of 500 ml / min / m2 as a reference? If ICU patients normally have lower values?

Re.: The reviewer comments on an important issue regarding our chosen reference point for the analysis. Of course, we fully agree that critically ill patients often present with a low DO2I before and at admission on the ICU which we were not able to monitor reliably. It is reasonable to assume that this insufficient oxygen delivery might trigger pathophysiological mechanisms causing intestinal ischemia which is then detected at ICU. Because of this we choose to exclude patients with II manifesting in the first 72 hours of intensive care treatment. As a DO2I of 500ml/min/m² is commonly reported as reference value in healthy subjects and typically not associated with intestinal ischemia and post procedural complications it was chosen as a reference value [9, 10]. On top of that a DO2I of 500ml/min/m2 was tested as a safe endpoint for shock resuscitation [11]. We commented on this in the Discussion section (page 15, lines 331-334).

11. Regarding the lactate analysis, they have higher levels in the control group, if the n of each group are considered, for which I do not consider the ROC analysis they present adequate. Why not use venous oxygen saturation which reflects systemic oxygenation?

Re.: The reviewer comments on the diagnostic value of lactate which we agree is an important fact. Regarding the apprehension of an uneven distribution of the lactate levels, please find enclosed in Fig 1 the lactate levels incorporated in the ROC analysis in the manuscript (control group: 3.5 (2.3/6.4) mmol/l vs. intestinal ischemia: 4.5 (2.3/7.3) mmol/l, p = 0.345). Furthermore, we now included ROC analyses of pH, PaCO2 and ScvO2 in the manuscript (page 12, lines 270-276 and Fig 4).

Fig 1.: Lactate levels

12. There are studies reported in critically ill patients with ischemia showing that lactate is not a good marker. The same authors report some references related to its low sensitivity and specificity. So why did they use it to predict II?

Re.: We fully agree with the reviewer on the limited clinical value of lactate as a predictive marker on intestinal ischemia in a relevant part of the literature. Nevertheless, as it was the aim of this study to define a delivery of oxygen resulting in II. It is undisputed, that lactate is an important metabolic marker indicating parenchymal ischemia [12] and various shock states [13, 14]. Furthermore, the measurement of lactate is recommended in current guidelines for intestinal ischemia [5, 7]. We therefore wanted to connote our findings of a “cut-off” DO2I with the corresponding lactate levels. Because of this our findings of relative high mean and minDO2I associated with II might help to explain the relative low sensitivity and specificity of lactate in the diagnosis of II. We elaborated on this in the discussion (page 17, lines 377-385).

13. I believe that the discussion should better relate the results of this work with what was previously reported. Why not analyze gastrointestinal ischemia or ischemia / reperfusion studies and compare their findings if they are closely related to what they are presenting? Many of the ICU patients develop sepsis, did they record the comorbidities of the patients included in the study?

Re.: According to the recommendations of the reviewer we restructured the discussion (pages 13-18, lines 296-418)

Literature: 

1. Al-Diery H, Phillips A, Evennett N, Pandanaboyana S, Gilham M, Windsor JA. The Pathogenesis of Nonocclusive Mesenteric Ischemia: Implications for Research and Clinical Practice. J Intensive Care Med. 2019;34(10):771-81. Epub 2018/07/25. doi: 10.1177/0885066618788827. 

2. Trompeter M, Brazda T, Remy CT, Vestring T, Reimer P. Non-occlusive mesenteric ischemia: etiology, diagnosis, and interventional therapy. Eur Radiol. 2002;12(5):1179-87. Epub 2002/04/27. doi: 10.1007/s00330-001-1220-2

3. Acosta S, Bjorck M. Acute thrombo-embolic occlusion of the superior mesenteric artery: a prospective study in a well defined population. Eur J Vasc Endovasc Surg. 2003;26(2):179-83. Epub 2003/08/15. doi: 10.1053/ejvs.2002.1893

4. Duran M, Pohl E, Grabitz K, Schelzig H, Sagban TA, Simon F. The importance of open emergency surgery in the treatment of acute mesenteric ischemia. World J Emerg Surg. 2015;10:45. Epub 2015/09/29. doi: 10.1186/s13017-015-0041-6.

5. Bala M, Kashuk J, Moore EE, Kluger Y, Biffl W, Gomes CA, et al. Acute mesenteric ischemia: guidelines of the World Society of Emergency Surgery. World J Emerg Surg. 2017;12:38. Epub 2017/08/11. doi: 10.1186/s13017-017-0150-5

6. Oldenburg WA, Lau LL, Rodenberg TJ, Edmonds HJ, Burger CD. Acute mesenteric ischemia: a clinical review. Arch Intern Med. 2004;164(10):1054-62. Epub 2004/05/26. doi: 10.1001/archinte.164.10.1054

7. Tilsed JV, Casamassima A, Kurihara H, Mariani D, Martinez I, Pereira J, et al. ESTES guidelines: acute mesenteric ischaemia. Eur J Trauma Emerg Surg. 2016;42(2):253-70. Epub 2016/01/29. doi: 10.1007/s00068-016-0634-0

8. Eilers PH MB. Flexible smoothing with B-splines and penalties. Statistical Science. 1996;11:89-121.

9. Futier E, Robin E, Jabaudon M, Guerin R, Petit A, Bazin JE, et al. Central venous O(2) saturation and venous-to-arterial CO(2) difference as complementary tools for goal-directed therapy during high-risk surgery. Crit Care. 2010;14(5):R193. Epub 2010/11/03. doi: 10.1186/cc9310

10. Bartha E, Arfwedson C, Imnell A, Kalman S. Towards individualized perioperative, goal-directed haemodynamic algorithms for patients of advanced age: observations during a randomized controlled trial (NCT01141894). Br J Anaesth. 2016;116(4):486-92. Epub 2016/03/20. doi: 10.1093/bja/aew025

11. McKinley BA, Kozar RA, Cocanour CS, Valdivia A, Sailors RM, Ware DN, et al. Normal versus supranormal oxygen delivery goals in shock resuscitation: the response is the same. J Trauma. 2002;53(5):825-32. Epub 2002/11/19. doi: 10.1097/00005373-200211000-00004

12. Nuzzo A, Maggiori L, Ronot M, Becq A, Plessier A, Gault N, et al. Predictive Factors of Intestinal Necrosis in Acute Mesenteric Ischemia: Prospective Study from an Intestinal Stroke Center. Am J Gastroenterol. 2017;112(4):597-605. Epub 2017/03/08. doi: 10.1038/ajg.2017.38

13. Hernandez G, Ospina-Tascon GA, Damiani LP, Estenssoro E, Dubin A, Hurtado J, et al. Effect of a Resuscitation Strategy Targeting Peripheral Perfusion Status vs Serum Lactate Levels on 28-Day Mortality Among Patients With Septic Shock: The ANDROMEDA-SHOCK Randomized Clinical Trial. JAMA. 2019;321(7):654-64. Epub 2019/02/18. doi: 10.1001/jama.2019.0071

14. Frydland M, Moller JE, Wiberg S, Lindholm MG, Hansen R, Henriques JPS, et al. Lactate is a Prognostic Factor in Patients Admitted With Suspected ST-Elevation Myocardial Infarction. Shock. 2019;51(3):321-7. Epub 2018/10/05. doi: 10.1097/SHK.0000000000001191

---

## [Editor Report · Decision Letter 1]

25 Jun 2021

Maintaining oxygen delivery is crucial to prevent intestinal ischemia in critical ill patients

PONE-D-21-08068R1

Dear Dr. Schöttler,

We’re pleased to inform you that your manuscript has been judged scientifically suitable for publication and will be formally accepted for publication once it meets all outstanding technical requirements.

Kind regards,

Corstiaan den Uil

Academic Editor

PLOS ONE
---

## [Editor Report · Acceptance letter]

1 Jul 2021

PONE-D-21-08068R1 

Maintaining oxygen delivery is crucial to prevent intestinal ischemia in critical ill patients 

Dear Dr. Schoettler:

I'm pleased to inform you that your manuscript has been deemed suitable for publication in PLOS ONE. Congratulations! Your manuscript is now with our production department. 

Kind regards, 

on behalf of

Dr. Corstiaan den Uil 

Academic Editor

PLOS ONE